

**A High-Precision Satellite XCH₄ Inversion Method Using CBAM-ResNet18**
Lu Fan,[1] Yong Wan,[2,*] Yuyu Chen,[2]  Yongshou Dai, [2] Shaokun Xu [2]
[1] Technical Testing Center of Shengli Oilfield Branch, China Petroleum & Chemical Corporation, Dong ying
257000; fanlu550.slyt@sinopec.com (L.F.)
[2] College of Oceanography and Space Informatics, China University of Petroleum (East China), Qingdao,
Shandong 266580,China; wanyong@upc.edu.cn(Y.W.); c17863959943@163.com (Y.C); daiys@upc.edu.cn
(Y.D.); 1753970610@qq.com (S.X.).
*Correspondence to: wanyong@upc.edu.cn(Y.W.)*
**Abstract**. Amid global climate change, rising atmospheric methane (CH₄) concentrations signif-icantly influence the
climate system, contributing to temperature increases and at-mospheric chemistry changes. Accurate monitoring of
these concentrations is essential to support global methane emission reduction goals, such as those outlined in the
Global Methane Pledge targeting a 30% reduction by 2030. Satellite remote sensing, offering high precision and
extensive spatial coverage, has become a critical tool for measuring large-scale atmospheric methane concentrations.
However, traditional physical inversion models face challenges, including high computational complexity, low
processing efficiency, and inadequate incorporation of spatial distribution infor-mation, limiting their effectiveness.
To address these shortcomings, this study proposes a high-precision XCH₄ inversion method that integrates the
Convolutional Block At-tention Module (CBAM) with the ResNet18 neural network (CBAM-ResNet18). By
leveraging shortwave infrared spectral data from the Sentinel-5P satellite and the CAMS reanalysis dataset, this
approach achieves rapid and accurate XCH₄ inversion. Experimental results demonstrate that the method outperforms
both conventional physical models and existing mainstream techniques in terms of inversion accuracy and
computational efficiency. It achieves an error of less than 2%, meeting the strin-gent precision requirements for XCH₄
in atmospheric remote sensing and providing a robust tool for methane monitoring.

**Keywords**: XCH₄ inversion, satellite remote sensing, CBAM-ResNet18, methane monitoring

**1   Introduction**

Amid global climate change, methane (CH₄), a potent greenhouse gas, has signifi-cantly influenced the climate

through rising atmospheric concentrations, driving tem-perature increases and altering atmospheric chemistry (Thakur
& Solanki, 2022; Winterstein et al., 2019). Since the Industrial Revolution, extensive fossil fuel use, expanded
agricultural activities, and waste manage-ment practices have markedly elevated methane levels (Wuebbles & Hayhoe,
2002, Hinrich, 2019). As a result, the climate impacts of methane emissions and concentration changes have gained
growing atten-tion. At the 26th United Nations Climate Change Conference (COP26), over 100 coun-tries committed
to the Global Methane Pledge, aiming to reduce global methane emis-sions by at least 30% by 2030 (Vogel, 2022).
To achieve this goal, scientists have recommended enhancing methane emission monitoring and modeling capabilities,
including im-proving process models, expanding wetland flux measurements, extending fossil fuel emission



measurements, and refining data from waste management systems (Anita et al., 2019). These advancements will
facilitate more accurate quantification of methane emissions and the formulation of effective mitigation strategies.
Among these efforts, effectively monitoring changes in methane concentration serves as the foundation for various
mitigation initiatives. It provides critical information on methane concentration varia-tions and offers precise
quantitative references for methane reduction measures (Erland et al., 2022). Therefore, effective monitoring of
atmospheric methane concentration is essential for mitigating climate change.
To effectively monitor and assess methane emissions, satellite remote sensing technology has become a crucial
tool for obtaining large-scale atmospheric methane concentrations, leveraging its advantages of high precision and
extensive monitoring coverage (Jacob et al., 2022). Methane monitoring satellites typically utilize thermal infrared or
shortwave infrared hyperspectral sensors to capture spectral information reflected from the Earth's surface. Through
inversion algorithms, this spectral information is then converted into $XCH_4$ data (Worden et al., 2015). $XCH_4$, an
essential metric derived from satellite observations, represents the column-averaged dry-air mole fraction of methane
and has been widely used for quantifying atmospheric methane concentrations (Zeng et al., 2021).
Early research on $XCH_4$ inversion methods primarily relied on radiative transfer models, such as the widely used
MODTRAN (MODerate resolution atmospheric TRANsmission) and LBLRTM (Line-By-Line Radiative Transfer
Model) (Rothman et al., 2017; Clough et al., 2005). These models invert $XCH_4$ by performing precise calculations of
atmospheric spectra in con-junction with satellite observation data, such as those from TROPOMI or GOSAT. In
recent years, the CAMS (Copernicus Atmosphere Monitoring Service) reanalysis data provided by the European
Centre for Medium-Range Weather Forecasts (ECMWF) has also offered richer reference information for atmospheric
methane concentration in-version (Inness et al., 2019). However, most current $XCH_4$ inversion methods depend on
physics-based inversion algorithms, which often face challenges such as high computational com-plexity and slow
processing speeds when handling satellite data. Additionally, these methods predominantly rely on single-point
detection data and fail to fully utilize spa-tial distribution information, thereby compromising the accuracy and
efficiency of $XCH_4$ inversion (Jacob et al., 2019; Pandey et al., 2022).
In recent years, with the rapid development of deep learning techniques, da-ta-driven methods based on neural
networks have demonstrated significant ad-vantages in remote sensing data analysis (Zhu et al., 2017). Among these,
the Residual Neural Net-work (ResNet) has been widely applied in fields such as image classification and object
detection due to its powerful feature extraction capabilities in deep layers (He et al., 2016). The Convolutional Block
Attention Module (CBAM), an attention mechanism designed to enhance the performance of convolutional neural
networks (CNNs), selectively focus-es on important feature channels and spatial locations during the feature extraction
process by integrating channel attention and spatial attention modules, thereby im-proving the network's ability to
perceive critical information (Praharsha & Poulose, 2024). Currently, CBAM has been extensively utilized in various
remote sensing applications, including su-per-resolution reconstruction (Wang et al., 2024), change detection (Wang
et al., 2022), image segmentation (Shun et al., 2022), and image fusion (Liu et al., 2023). Research results indicate
that by incorporating CBAM, the quality and processing accuracy of remote sensing images have been significantly
im-proved, highlighting the immense potential of CBAM in the field of remote sensing.



To further enhance the efficiency and accuracy of XCH₄ inversion, this study pro-poses an XCH₄ inversion
method based on the CBAM attention mechanism and the ResNet18 neural network, utilizing shortwave infrared
spectral data from the Senti-nel-5P satellite and the CAMS reanalysis data from the European Centre for Medi-um-
Range Weather Forecasts (ECMWF). By integrating spatial distribution infor-mation and spectral features, this
method significantly improves inversion accuracy and accelerates computational speed, achieving the goal of rapidly
and accurately ob-taining atmospheric methane concentrations from satellite data.
The structure of this paper is organized as follows: Section 2 provides a detailed introduction to the sources and
preprocessing methods of the satellite data and me-thane concentration data used in this study, as well as the
construction process of the CBAM-ResNet18 model and its application in XCH₄ inversion. Section 3 presents ex-
perimental visualizations, validation, and analytical discussions. Finally, Section 4 summarizes the research findings
of this study.
for the reviewers. The final layout of the typeset paper will not match this template layout.
**2    Materials and Methods**
*2.1   Data preprocessing*
High-precision XCH₄ inversion requires high-quality data samples. This section outlines the preprocessing
steps for Sentinel-5P L1B spectral data and CAMS XCH₄ reanalysis data.
*2.1.1 Preprocessing of Satellite L1B Spectral Data*
To extract high-quality samples from satellite observation data, this study begins with the L1B-level spectral data
from the Sentinel-5P satellite. The primary data pre-processing steps consist of three parts: data filtering, spatial
cropping, and spectral data normalization.
1.  Data Filtering: Since the L2 products have undergone certain data filtering pro-cesses (e.g., quality control, cloud
masking, etc.), the L1B data used in the genera-tion of L2 products can be considered of relatively high quality.
Therefore, based on the valid detection pixels of the L2 products, we extracted the corresponding L1B spectral
data. Observations impacted by clouds or high aerosol optical depth (AOD) were excluded. Clouds significantly
impair satellite detection of surface reflectance, while high AOD disrupts spectral signals. Thus, using the cloud
mask and aerosol optical depth parameters from the L2 products, we set filtering thresholds to retain only
observation pixels with cloud fraction below 0.1 and AOD below 0.2.
2.  Spatial Cropping: To leverage the spatial distribution information of methane, we cropped the satellite observation
data into 3×3 data blocks. Each block contains three pixels in both the longitude and latitude directions, resulting
in data blocks with spatial dimensions, thereby enhancing the model's ability to capture spatial correlations. The
corresponding L1B spectral data for each block forms a 3×3×480 three-dimensional tensor during inversion,
where 480 represents the number of channels in the L1B spectrum, as illustrated in Figure 1. This cropping method
allows the model to capture the distribution patterns of methane concentrations within localized spatial ranges,
improving sensitivity to spatial heterogeneity.





**Fig. 1** Schematic Diagram of Spatial Cropping.
3. Spectral Data Normalization: To eliminate scale differences among spectral data from different bands, this study
normalized the spectral data for each pixel. Assuming the original spectral data is I(λ), the normalization is
performed for each channel λ using the following formula:

$$I'(\lambda) = \frac{I(\lambda) - \mu_\lambda}{\sigma_\lambda}, \tag{1}$$

where $\mu_\lambda$ and $\sigma_\lambda$ represent the mean and standard deviation of channel λ, respectively. Normalization helps
accelerate the model training process and prevents issues such as gradient vanishing or explosion
*2.1.2 Preprocessing of CAMS XCH₄ Reanalysis Data*
This study conducted spatiotemporal matching and interpolation on the CAMS XCH₄ reanalysis data. To
effectively correlate the satellite-observed spectral data with surface XCH₄, the CAMS reanalysis data were used for
spatiotemporal matching. Specifically, for each satellite observation time and location, the CAMS data were linearly
interpolated to ensure consistency in both temporal and spatial dimensions. Additionally, normalization of the dataset
was performed to eliminate scale differences across different dimensions, thereby enhancing the training efficiency of
the model and the accuracy of the inversion results.
When using deep learning models for inversion, the spatiotemporal consistency between training data and target
data is crucial for ensuring model performance. To guarantee the temporal and spatial alignment between Sentinel-5P
observation data and CAMS XCH₄ data, the following processing steps were implemented in this study:
1. Spatiotemporal Matching: The CAMS XCH₄ data has a temporal resolution of 3 hours and a spatial
resolution of 0.75° × 0.75°. To align it with Sentinel-5P observation data, the temporal and spatial resolution
of the CAMS data was first adjusted to match that of the satellite data. For temporal matching, assuming a
satellite observation time $t_s$, and $t_1$ and $t_2$ as the two closest time points provided by CAMS (i.e., $t_1 < t_s <$
$t_2$)), the XCH₄ at time $t_s$ is calculated through linear interpolation:.

$$X_{CH_4}(t_s) = X_{CH_4}(t_1) + \frac{t_s - t_1}{t_2 - t_1} \cdot (X_{CH_4}(t_2) - X_{CH_4}(t_1)) \tag{2}$$

where $X_{CH_4}(t_1)$ and $X_{CH_4}(t_2)$ represent the XCH₄ values from CAMS at times $t_1$ and $t_2$, respectively.

2. Spatial Interpolation: In the spatial dimension, since the resolution of CAMS data is 0.75°, while Sentinel-5P
observations have a higher resolution (approximately 5.5 km × 7 km), this study employed bilinear





interpolation to align the CAMS data with the geographic locations of the satellite observation points. For a satellite observation point with longitude $\lambda_s$ and latitude $\phi_s$, the methane concentration values at the four nearest CAMS grid points are denoted as X11, X12, X21, and X22. The concentration value at the observation point is calculated using bilinear interpolation as follows:

$$X_{CH_4(\lambda_s,\phi_s)} = \frac{1}{(\lambda_2-\lambda_1)(\phi_2-\phi_1)}[X_{11}(\lambda_2-\lambda_s)(\phi_2-\phi_s) + X_{12}(\lambda_s-\lambda_1)(\phi_2-\phi_s) + X_{21}(\lambda_s - \lambda_1)(\phi_2-\phi_s) + X_{22}(\lambda_s-\lambda_1)(\phi_s-\phi_1)], \tag{3}$$

where $\lambda_1$, $\lambda_2$ and $\phi_1$, $\phi_2$ represent the longitude and latitude boundaries of the CAMS grid, respectively. Through this process, the XCH$_4$ data from CAMS are precisely mapped to the observation locations of Sentinel-5P.

3. Interpolation Error Control: To control potential errors introduced during the interpolation process, this study compared the interpolated CAMS data with the original resolution observations after interpolation, ensuring that the interpolation error remained within acceptable limits. Specifically, the Root Mean Square Error (RMSE) was used to evaluate the interpolation quality, calculated as follows:

$$RMSE = \sqrt{\frac{1}{N}\sum_{i=1}^{N}(X_{CH_4}^{interpolated} - X_{CH_4}^{original})^2}, \tag{4}$$

where N is the total number of interpolated data points, and $X_{CH_4}^{interpolated}$ interpolated and $X_{CH_4}^{original}$ original represent the interpolated XCH$_4$ and the original data, respectively.

Through these processing steps, this study effectively ensured precise spatiotemporal alignment between Sentinel-5P observation data and CAMS reanalysis data, forming high-quality input-output pairs for the CBAM-ResNet18 model to train and perform inversions.

*2.2 High-Precision XCH$_4$ Inversion Method*

*2.2.1 CBAM-ResNet18*

Convolutional neural networks (CNNs) excel at extracting high-dimensional features, with their expressive power and feature extraction capabilities enhancing as network depth increases. However, merely adding layers can cause performance degradation. ResNet18 addresses this by introducing skip connections to optimize the neural network architecture, combined with batch normalization, and eliminates the traditional fully connected layer at the end. The core of deep residual networks is the residual unit, as shown in Figure 2. In traditional neural network structures, it is often difficult to directly achieve an identity mapping where the output is identical to the input (i.e., H(x)=x). Residual neural networks, however, allow the residual block to focus on learning the residual value F(x)=H(x)-x. When the residual F(x) equals zero, it effectively constructs an identity mapping. Compared to directly learning identity mappings, this simplifies the learning task and reduces its difficulty. By adopting the residual learning mechanism, deep residual networks effectively mitigate the performance degradation issue that arises when stacking layers in deep CNNs, theoretically allowing unlimited increases in network depth to enhance model prediction accuracy. To ensure both the accuracy and real-time performance of XCH$_4$ inversion, ResNet18, with its moderate depth and faster convergence, was selected as the training model.




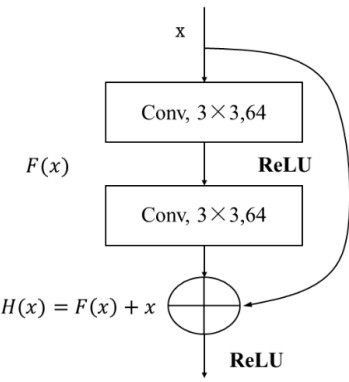

**Fig. 2** Schematic Diagram of the Residual Unit.
CBAM (Convolutional Block Attention Module) is a lightweight convolutional attention module that consists of
two sub-modules: CAM (Channel Attention Module) and SAM (Spatial Attention Module), which perform channel
attention and spatial attention, respectively. It can be integrated as a plug-and-play module into existing network
architectures. For an input feature $F \in R^{C*H*W}$, the channel attention module applies a 1D convolution $Mc \in R^{C*}$
$^{1*1}$, multiplies the convolution result with the original feature map, and uses the CAM output as the input for the
spatial attention module, which applies a 2D convolution $Ms \in R^{1*H*W}$. The final output feature is obtained by
multiplying the result with the original feature map. The structure of the CBAM attention mechanism is illustrated in
Figure 3.

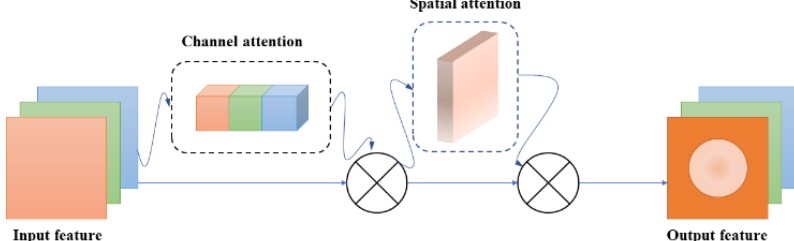

**Fig. 3** Schematic Diagram of the CBAM Attention Mechanism.
In this study, the CBAM module was integrated into the ResNet18 network structure, placed after the residual
module. In the CBAM-ResNet18 architecture, the input data first passes through the residual module for feature
extraction. The extracted features then enter the CBAM module for attention adjustment across spatial and channel
dimensions, generating weights that represent the importance of spatial and channel features. These weights are used
to amplify or reduce the original feature map accordingly, enabling a deep, multi-dimensional understanding and
optimization of features throughout the network, thereby enhancing the performance of ResNet18. The main modules
of CBAM-ResNet18 are illustrated in Figure 4.



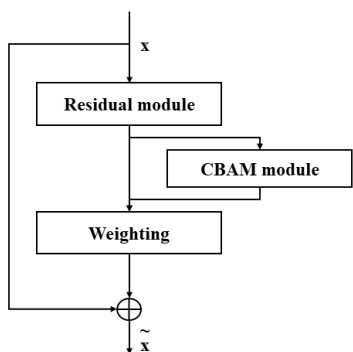

**Fig. 4** Schematic Diagram of the Main Modules of CBAM-ResNet18.
*2.2.2 Establishment of XCH₄ Inversion Method Based on CBAM-ResNet18*
The input data to the model takes the form of $n \times 3 \times 3 \times 480$ (where n is the number of samples), and the model
output is the XCH₄ data provided by CAMS, which is spatiotemporally matched with the input data. The training set
consists of 2244 samples, and the test set contains 647 samples. During the model training process, input parameters
are normalized to eliminate the influence of different feature scales and improve training efficiency. This is essentially
a regression task. Therefore, in the ResNet18 model structure, the original softmax layer is removed and replaced with
a combination of a global average pooling layer, a fully connected layer, and a regression layer. During the model
training phase, the loss function quantifies the difference between the network's actual output and the target result,
and this difference is backpropagated through the network to update and optimize all parameters of the neural network.
Initializing the parameters of a neural network is one of the critical steps in the deep learning training process.
The rationality of their settings directly affects whether the model can effectively learn and fit the training data. Batch
size refers to the size of the data subset used for each gradient update. A reasonable batch size can effectively balance
the model's convergence speed and optimization performance. While a smaller batch size may accelerate convergence,
it can also introduce more noise, making the training process less stable. The learning rate is also a crucial
hyperparameter. If the learning rate is too large, it may cause the model to oscillate during training, making it difficult
to converge. Conversely, if the learning rate is too small, the training process may become slow or even get trapped
in a local optimum. In this study, the model training employed the Adam adaptive optimization algorithm, with the
batch size set to 150 and the initial learning rate set to 0.01.
**3 EXPERIMENT AND ANALYSIS**
*3.1 Experimental environment*
The experiment was run in the following hardware environment: Intel Xeon(R) Gold 6226R@2.90GHz 2.89GHz
CPU, 256GB RAM, NVIDIA Quadro RTX 6000 GPU, Windows 10 Professional Edition, MATLAB R2022a.





*3.2 Experimental environment*
*3.2.1 Sentinel-5P satellite data*
The European Space Agency launched the Sentinel-5P satellite, equipped with the Tropospheric Monitoring
Instrument (TROPOMI), into space on October 13, 2017. As a satellite deployed in a near-polar sun-synchronous
orbit, Sentinel-5P undertakes the critical mission of atmospheric composition monitoring. Its primary goal is to
achieve high spatiotemporal resolution remote sensing, accurately measuring key components of Earth's atmosphere,
such as methane ($CH_4$), nitrogen dioxide ($NO_2$), ozone ($O_3$), and aerosols, as well as monitoring ultraviolet radiation
intensity. These data are of significant importance for air quality assessment, climate change research, and the
improvement of climate prediction models. Leveraging its unique orbital characteristics, Sentinel-5P provides a wide
swath imaging capability of approximately 2600 kilometers, enabling daily global coverage. The satellite completes
a precise global revisit every 16 days. The spectrometer onboard TROPOMI performs push-broom observations across
multiple spectral bands, covering seven bands from ultraviolet to shortwave infrared. In the shortwave infrared band,
the spatial resolution can reach 5.5 km × 7 km. The Sentinel-5P official website distributes Level 1B and Level 2
products to users. Level 1B products consist of spectral data, while Level 2 products include $XCH_4$, XCO, and cloud
mask data, among others. This study will focus on the inversion of $XCH_4$ based on the L1B_RA_BD7 spectral data,
combined with aerosol optical depth and cloud mask parameters from the Level 2 $XCH_4$ product.
*3.2.2 CAMS Reanalysis Data*
The model output utilizes $XCH_4$ reanalysis data provided by CAMS (Copernicus Atmosphere Monitoring
Service). The CAMS reanalysis dataset includes estimates of greenhouse gases and other variables from 2003 to 2020,
with a temporal resolution of 3 hours and a spatial resolution of 0.75° × 0.75°.
*3.2.3 TCCON Site Data*
TCCON (Total Carbon Column Observing Network) is a global ground-based ob-servation network that uses
Fourier Transform Spectroscopy (FTS) to measure spectral data in the near-infrared band of solar radiation. By
applying a nonlinear least squares fitting algorithm, TCCON precisely retrieves the column-averaged dry-air mole
frac-tions of atmospheric components, such as $CH_4$ ($XCH_4$), from the observed spectral data. TCCON sites exhibit
extremely high retrieval accuracy, and all site data are inde-pendently validated, ensuring their reliability. Using
TCCON site observation data to validate inversion results is currently a mainstream practice.
To facilitate the validation of inversion results, 15 orbital data sets covering North and South America between
June 1, 2020, and June 15, 2020, were selected. This region was chosen due to its higher density of TCCON sites.
*3.3 XCH₄ Inversion Results and Validation Analysis*
To intuitively display and validate the inversion results from multiple perspectives, this section selects 15
Sentinel-5P orbital spectral datasets covering North and South America between June 1, 2020, and June 15, 2020, for



inversion. The results are presented in the form of spatial distribution maps and compared with XCH₄ reanalysis data
from CAMS, observational data from TCCON sites, and inversion results from other mainstream methods.
*3.3.1 Visualization of XCH₄ Inversion Results*
To more clearly and intuitively observe the spatial distribution characteristics of the inversion results, this section
utilizes the CBAM-ResNet18 inversion model to conduct XCH₄ concentration inversion for the complete regions of
Oklahoma on June 11, 2020, and Wisconsin on June 14, 2020. The spatial distribution maps of the inversion results
are generated and shown in Figure 5. These maps illustrate the variations in XCH₄ within the selected areas. The XCH₄
data processed by the CBAM-ResNet18 model can reflect the heterogeneity of methane concentrations across different
geographical locations, such as urban areas, industrial zones, and natural wetlands, which are potential sources of
methane emissions.


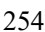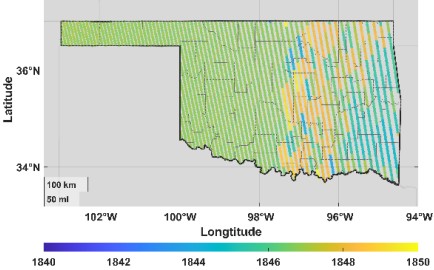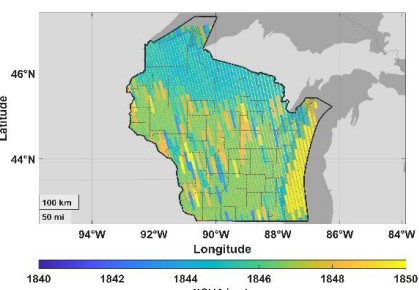

**(a)** Spatial Distribution Map of Inversion Results for Oklahoma on June 11, 2020

**(b)** Spatial Distribution Map of Inversion Results for Wisconsin on June 14, 2020

**Fig. 5** Spatial Distribution Maps of XCH₄ Inversion Results from the CBAM-ResNet18 Model.

Figure 5(a) reveals a pronounced spatial gradient in XCH₄ concentrations, rising from 1840–1842 ppb in the west
(102°W to 100°W) to 1848–1850 ppb in the east (98°W to 94°W). The model successfully captures this west-to-east
concentration gradient, demonstrating its strong spatial resolution capability over large spatial scales (state-level,
approximately 400 km × 600 km). Additionally, the concentration changes across grid cells appear smooth, without
noticeable abrupt transitions or noise, indicating the stable performance of the inversion method across different
locations. In Figure 5(b), the XCH₄ concentration increases from lower values (1840-1844 ppb) in the west (94°W to
90°W) to higher values (approaching 1850 ppb) in the east (88°W to 84°W). The model also captures this spatial
gradient, particularly in the eastern region near Lake Michigan, where the concentration changes are more nuanced
and transitions between grid cells are natural. This suggests that the inversion method effectively resolves the spatial
distribution characteristics of XCH₄ concentration across different geographical environments (the flat terrain of
Oklahoma versus the lake and forest terrain of Wisconsin).



From Figure 5, it can be observed that the XCH₄ concentration range (1840-1850 ppb) in both maps indicates
that the model exhibits high sensitivity to small-scale concentration variations (10 ppb). For example, in eastern
Oklahoma, the model can distinguish subtle differences between localized high-value areas (1848-1850 ppb) and
surrounding regions (1846-1848 ppb). Similarly, in eastern Wisconsin, the model captures the concentration peak near
Lake Michigan. This sensitivity demonstrates that the inversion method can effectively extract minor variations in
XCH₄ concentration when processing high-resolution satellite data, making it suitable for identifying potential
methane emission hotspots.
The above visualization results preliminarily indicate that the CBAM-ResNet18 model performs well in XCH₄
inversion using satellite spectral data and spatial information. Additionally, the resolution and detail retention of the
inversion results benefit from the model's ability to extract spatial features from 3×3 data blocks, enabling effective
identification of local methane concentration variations.
*3.3.2 Visualization of XCH₄ Inversion Results*
This study employs Root Mean Square Error (RMSE) and Mean Absolute Error (MAE) to evaluate the accuracy
of the inverted XCH₄. The formulas for the relevant evaluation metrics are as follows:

$$RMSE = \sqrt{\frac{1}{N}\sum_{i=1}^{N}(f_i - y_i)^2}., \tag{5}$$

$$MAE = \frac{1}{N}\sum_{i=1}^{N}|f_i - y_i|, \tag{6}$$

where N is the number of samples, $f_i$ is the predicted value, and $y_i$ is the true value.
Using the proposed XCH₄ inversion method, Sentinel-5P satellite observation data were inverted to obtain XCH₄.
This study first validated the inversion results using CAMS reanalysis data. The validation results are shown in Figure

6.

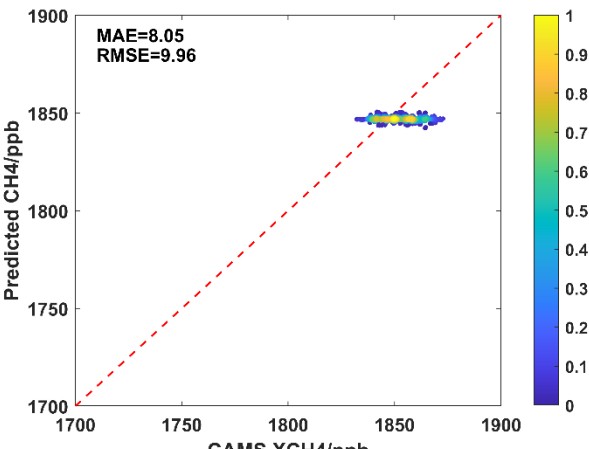

**Fig. 6** Comparison of Inversion Results and CAMS Reanalysis Data.





From the scatter plot comparing the CBAM-ResNet18 inversion results with CAMS reanalysis data in Figure 6,
it can be observed that the Mean Absolute Error (MAE) is 8.05 ppb, and the Root Mean Square Error (RMSE) is 9.96
ppb. This demonstrates that the proposed CBAM-ResNet18-based inversion method for Sentinel-5P satellite
observation data meets the requirement of XCH$_4$ accuracy being less than 2% in the field of atmospheric remote
sensing.
*3.3.3 Validation of Inversion Results Based on TCCON Site Observation Data*
To further validate the accuracy of the inversion results, the inversion results were verified based on TCCON
observation data from two sites. The information of the TCCON sites used is shown in Table 1, and the validation
results are presented in Figure 6.

**Table 1.** TCCON site information.

| Number | Site | Longitude / °W | Latitude / °N |
|---|---|---|---|
| 1 | Lamont (US) | -97.49 | 36.6 |
| 2 | Park Falls (US) | -90.27 | 45.94 |


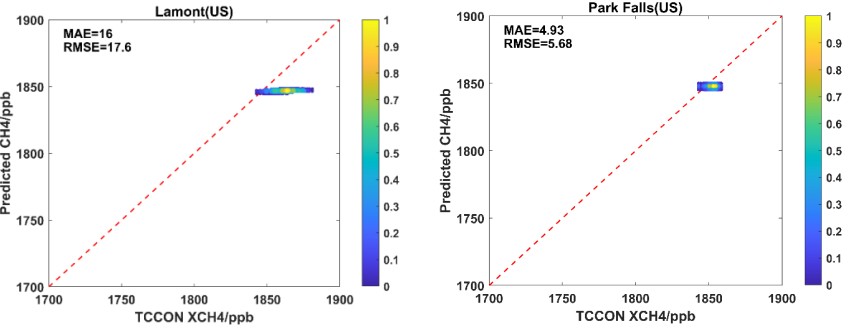


**(a)** Validation Results of Inversion Based on Lamont (US) Site    **(b)** Validation Results of Inversion Based on Park Falls (US) Site

**Figure 7.** Comparison of Inversion Results and TCCON Site Observation Data.
From the scatter plots in Figure 7, which compare the CBAM-ResNet18 inversion results with ground-based site
observation data, it can be observed that:
1.  The Mean Absolute Error (MAE) between the CBAM-ResNet18 inversion results and the XCH$_4$
observations from the Lamont (US) site is 16 ppb, and the Root Mean Square Error (RMSE) is 17.6 ppb.
2.  The MAE between the CBAM-ResNet18 inversion results and the XCH$_4$ observations from the Park Falls
(US) site is 4.93 ppb, and the RMSE is 5.68 ppb.
Based on these statistical parameters, it is evident that the proposed CBAM-ResNet18-based inversion method
for Sentinel-5P satellite observation data meets the requirement of XCH$_4$ accuracy being less than 2% in the field of
atmospheric remote sensing.



*3.3.4 Comparative Analysis with Mainstream Methods*
To comprehensively evaluate the performance of the proposed CBAM-ResNet18 model in $XCH_4$ inversion, this
section compares it with ResNet18, an improved spatial inversion method (Chen, 2023), the optimal estimation
method, and the $XCH_4$ data product from Sentinel-5P satellite (European Space Agency, 2021). The experiment aims
to demonstrate the advantages of the CBAM-ResNet18 model in terms of inversion accuracy and computational
efficiency through quantitative and qualitative analysis.
1.   Comparison of Inversion Accuracy
This comparative experiment involves the following five methods, which are compared with the CBAM-
ResNet18 model. The specific information is shown in Table 2.

**Table 2.** overview of comparative methods.

| Method | Description |
| --- | --- |
| CBAM-ResNet18 | Based on ResNet18, the CBAM attention mechanism is integrated to enhance feature extraction capabilities through spatial and channel attention. |
| ResNet18 | The baseline ResNet18 model, without CBAM, is used to evaluate the improvement brought by the attention mechanism. |
| Improved spatial inversion | An improved spatial inversion method that enhances spatial resolution and accuracy by optimizing spatial interpolation algorithms. |
| Optimal estimation | Based on Bayesian theory, the optimal inversion results are calculated by weighting prior information and observational data. |
| Sentinel-5P | The $XCH_4$ data product provided by the European Space Agency's Sentinel-5P satellite. |

To quantitatively evaluate the inversion accuracy of each method, the same evaluation metrics as before are used.
The CBAM-ResNet18 model is compared with the four methods at the Lamont (US) and Park Falls (US) sites,
respectively. The monitoring values from the two sites are used as reference values to calculate the average $XCH_4$
inversion accuracy of each method. The experimental results are shown in Table 3.

**Table 3.** compare of inversion accuracy among different method.

| Method | RMSE / ppb | MAE / ppb |
| --- | --- | --- |
| CBAM-ResNet18 | 11.64 | 10.465 |
| ResNet18 | 21.25 | 15.75 |
| Improved spatial inversion | 14.89 | 12.49 |
| Optimal estimation | 21.19 | 15.95 |
| Sentinel-5P | 17.18 | 13.93 |





From Table 3, it can be seen that the CBAM-ResNet18 model exhibits excellent average inversion accuracy at
both TCCON sites. Its average RMSE is 11.64 ppb, and its average MAE is 10.465 ppb, which is well below the
requirement of XCH$_4$ accuracy being less than 2% in the field of atmospheric remote sensing. This result indicates
that the CBAM-ResNet18 model can effectively utilize the spatial distribution information and spectral features of
Sentinel-5P satellite data, significantly improving the accuracy of XCH$_4$ inversion. Compared to the model using only
ResNet18, the introduction of CBAM significantly enhances inversion accuracy, demonstrating the crucial role of its
attention mechanism in feature extraction and spatial information fusion. Additionally, compared to the improved
spatial inversion method, the optimal estimation method, and the official Sentinel-5P XCH$_4$ product, the CBAM-
ResNet18 model also shows significant advantages in accuracy.
2.   Comparison of Computational Efficiency
Computational efficiency is an important metric for evaluating the practicality of inversion methods. Therefore,
the single inversion time of XCH$_4$ for the aforementioned five methods is compared, as shown in Table 4. The
computation time for Sentinel-5P is derived from the official algorithm documentation.

**Table 4.** compare of computational efficiency among different methods

| Method | Computation Time / s |
|---|---|
| CBAM-ResNet18 | 1.77 |
| ResNet18 | 1.65 |
| Improved spatial inversion | 1.95 |
| Optimal estimation | 7.62 |
| Sentinel-5P | 8.50 |

From Table 4, it can be observed that the CBAM-ResNet18 method also demonstrates significant advantages in
computational efficiency. Compared to traditional methods (such as the optimal estimation method and the method
used by Sentinel-5P), its computation time is reduced by approximately 76.8% and 79.2%, respectively. This
improvement is primarily attributed to the fast inference capability of the CBAM-ResNet18 neural network, which
directly derives XCH$_4$ data from satellite spectral data without the need for methane profile calculations required by
traditional methods. The improved spatial inversion method has a computation time of 1.95 seconds, slightly slower
than CBAM-ResNet18, which may be due to the additional computational overhead introduced by spatial correlation
calculations. However, compared to the baseline ResNet18 model, the computation time of CBAM-ResNet18 is
slightly increased (approximately 6.8%), likely due to the additional computational steps introduced by the CBAM
module.
In summary, the CBAM-ResNet18 model achieves a good balance between computational efficiency and model
complexity. While ensuring high inversion accuracy, it also reduces the computation time required for inversion,
enabling efficient and high-precision inversion of satellite spectral data to XCH$_4$.



## 4    CONCLUSION

This study presents a high-precision satellite XCH$_4$ inversion method using CBAM-ResNet18, integrating the CBAM attention mechanism with ResNet18 to deliver rapid, accurate atmospheric XCH$_4$ inversion, markedly enhancing both accuracy and efficiency. The method fully leverages the advantages of deep learning in feature extraction and spatial information fusion, providing an efficient and precise new approach for monitoring atmospheric methane concentrations. Experimental results demonstrate that the model exhibits high accuracy in comparative validations against CAMS reanalysis data and TCCON site data, with both Mean Absolute Error (MAE) and Root Mean Square Error (RMSE) within acceptable ranges, meeting the requirement of XCH$_4$ accuracy being less than 2% in the field of atmospheric remote sensing. Moreover, compared to ResNet18, the improved spatial inversion method, the optimal estimation method, and the official Sentinel-5P XCH$_4$ product, CBAM-ResNet18 shows significant advantages in inversion accuracy while also excelling in computational efficiency. The proposed method not only contributes to more accurate quantification of methane emissions but also provides essential data and technical support for global methane reduction efforts.

**Author Contributions**: Conceptualization, L.F., Y.D.; Methodology, Y.W., L.F.; Project administration, L.F., S.X., Y.D.; Software, L.F., Y.W.; Supervision, Y.W., Y.C., Y.D.; Validation, Y.C.; Visualization, S.X., L.F.; Writing—original draft, L.F., Y.C.; Writing—review and editing, L.F., S.X. All authors have read and agreed to the published version of the manuscript.

**Acknowledgments:** The authors would like to thank the anonymous reviewers for their suggestions.

**Conflicts of Interest:** The authors declare no conflicts of interest.

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
