# Peer review of "A High-Precision Satellite XCH4 Inversion Method Using CBAM-ResNet18"

_EGUsphere, 2025_

## Referee Comment (RC2)

**Review of: "A High-Precision Satellite XCH4 Inversion Method Using CBAM-ResNet18" by Lu Fan et al.**

**Overview**

This work presents a novel, data-driven XCH4 retrieval from Sentinel-5P employing the Convolutional Block Attention Module and the ResNet18 neural network. The topic is highly relevant and a lot of important advances in satellite retrieval algorithms were made in recent years with data-driven methods. In this case, however, I have significant doubts if the CAMS data set used for training of the neural network in this work is sufficient for this task (see major comments below). What is worse, I do not agree that the figures in the manuscript that represent the test results for the new retrieval method really support the claims that the authors make. In my understanding, they actually show significant issues with retrieval performance. Therefore, I cannot recommend this manuscript for publication.

**General/major comments**

- 1. Horizontal resolution of CAMS CAMS data set has much lower horizontal resolution than Sentinel-5P. Therefore, the neural network has no information on realistic small scale structure in methane concentration available to it, so I cannot see how such structure could be retrieved. The authors claim that the "spatial cropping" could somehow be used to address this issue, but they do not explain how this would work.
- 2. **Temporal interpolation** Another serious issue that would impact the resolution of the new retrievals is the temporal interpolation that the authors use. CAMS data is 3-hourly, hence the distance over which a small-scale methane structure could be advected in a single time step of the data set can be quite a bit larger than the size of a horizontal grid cell (the models themselves normally address this issue by running the dynamical core at much higher temporal resolution than the time step of model output). The authors use linear interpolation in time in this work, which would result in methane concentration structures being represented in incorrect locations (temporal "smearing" effect). This effect often causes problems when comparing models to observations, and would be even more relevant here, where a neural network is trained on a model.
- 3. The authors use Figures 5-7 to back their claims that the new retrieval method performs well. In my understanding, the Figure 5 is inconclusive at best, and Figures 6-7 appear to show poor performance of the retrieval. In particular:
  - (a) **Figure 5** The authors describe how the retrieval results shown in this figure reproduce various spatial features in the CAMS dataset. Therefore, I cannot understand why the corresponding CAMS data (for the same time and locations as the retrieval results) is not shown for comparison? Without it, it is not really possible to judge if the claims about the retrieval performance are correct.
  - (b) Figure 6-7 In my understanding, both figures show the range of variability in retrieved XCH4 which is much lower than the corresponding range in CAMS or TCCON data sets, which indicates very serious problems with the retrieval. The MAE and RMSE values reported may be lower than 2% when compared to the total amount of XCH4 observed, but they are about as large as the total variability of the retrieved signal (around 10 ppb, as represented in Figures 5-7), and therefore cannot be considered small or even acceptable. It is also strange that the figures contain a dashed red line, which would (correctly, in my understanding) indicate the expected position and distribution of data points for this type of comparison. The fact that the data points are not at all aligned with this line is seemingly not mentioned in the main text, this issue is not discussed in any way. In case I

have misinterpreted the data provided in some way, I apologize, but then the authors should really clarify what these figures represent.

4. Comparative analysis Section 3.3.4 presents comparative analysis with mainstream retrieval methods, but some of these, like optimal estimation, are very poorly described. While optimal estimation would probably be known (in general terms) to most people interested in this work, the details of its implementation are essential to judge wether it is a meaningful comparison to the results of this work.